# Evaluation and Application of Silk Fibroin Based Biomaterials to Promote Cartilage Regeneration in Osteoarthritis Therapy

**DOI:** 10.3390/biomedicines11082244

**Published:** 2023-08-10

**Authors:** Xudong Su, Li Wei, Zhenghao Xu, Leilei Qin, Jianye Yang, Yinshuang Zou, Chen Zhao, Li Chen, Ning Hu

**Affiliations:** 1Department of Orthopedics, The First Affiliated Hospital of Chongqing Medical University, Chongqing 400016, China; 2Laboratory of Orthopedics, Chongqing Medical University, Chongqing 400016, China

**Keywords:** silk fibroin, osteoarthritis, bone/cartilage repair, treatment

## Abstract

Osteoarthritis (OA) is a common joint disease characterized by cartilage damage and degeneration. Traditional treatments such as NSAIDs and joint replacement surgery only relieve pain and do not achieve complete cartilage regeneration. Silk fibroin (SF) biomaterials are novel materials that have been widely studied and applied to cartilage regeneration. By mimicking the fibrous structure and biological activity of collagen, SF biomaterials can promote the proliferation and differentiation of chondrocytes and contribute to the formation of new cartilage tissue. In addition, SF biomaterials have good biocompatibility and biodegradability and can be gradually absorbed and metabolized by the human body. Studies in recent years have shown that SF biomaterials have great potential in treating OA and show good clinical efficacy. Therefore, SF biomaterials are expected to be an effective treatment option for promoting cartilage regeneration and repair in patients with OA. This article provides an overview of the biological characteristics of SF, its role in bone and cartilage injuries, and its prospects in clinical applications to provide new perspectives and references for the field of bone and cartilage repair.

## 1. Introduction

Due to the gradual increase in the proportion of the aging population, obesity, and joint injuries worldwide, the risks of bone and cartilage diseases have significantly increased, seriously affecting people’s quality of life and physical health [1]. About 20~33% of the world’s population suffers from bone and joint diseases, and their incidence in adults is 30~40% [2], among which the incidence of osteoporosis [3], bone fractures [4], cartilage wear [5], and arthritis [6] gradually increases with age. Osteoarthritis (OA) is a progressive degenerative joint disease and the main cause of disability in adults, characterized by cartilage and subchondral bone degeneration [7]. Generally, cartilage and subchondral bone degeneration are due to increased metalloproteases and inflammatory cytokines; the excessive mechanical load and pathological factors cause bone and cartilage destruction, resulting in an imbalance in the dynamic equilibrium of repair and leading to OA. Thus, repairing damaged cartilage is a fundamental element in the treatment of OA [8,9].

Although clinical treatments for bone/cartilage-related diseases caused by OA include pain management, physical therapy, joint injection, and surgical treatment, they only improve the patients’ pain symptoms and do not achieve the goal of cartilage repair [10,11]. Autologous chondrocyte transplantation and allogeneic/autologous cartilage transplantation have also been applied in clinical practice, but there are still limitations, such as donor shortage, rejection reactions, and infections. Secondary surgery, limited collection sources, and complications often make tissue transplantation ineffective as a clinical treatment, especially for joint cartilage damage and key-sized bone defects [12,13,14,15]. Autologous and allogeneic transplantation are common clinical techniques for replacing damaged tissues, but they are limited by various factors, such as a lack of tissue that can be removed from healthy areas, and a lack of suitable donors. Allogeneic transplant material from donor tissue can cause immune responses, and, in cases of extensive injury and large surface areas, it is difficult to obtain appropriate materials on time, leading to low success rates [16]. Tissue engineering (TE) relies on the use of a variety of biocompatible materials to restore, maintain, and improve tissue function to regenerate injured tissues and organs. These materials can be seeded with cells and contain various supportive components. In recent years, TE has gained increasing attention as an alternative method for producing patient-specific tissues for repair and replacement applications [17,18,19,20], but various biomaterials have inherent limitations, so finding an excellent biomaterial has become the focus of research in recent decades [21].

Silk fibroin (SF) is a common natural material with excellent mechanical properties, low rejection reaction, tunable biodegradability, and good stability in the field of biomedical engineering, especially in tissue engineering [22,23,24]. SF is an important extracellular matrix protein, widely present in bone and cartilage tissues, and has important biological functions [25]. SF plays an important role in tissue engineering. Due to its good biocompatibility and biodegradability, SF can be used to build artificial tissues and organs. It can act as a scaffold or matrix material to promote cell attachment, proliferation, and differentiation in vivo and support new tissues. Secondly, SF also promotes wound healing. It has good biocompatibility and bioactivity to promote the regeneration and repair of traumatized tissue. SF can promote angiogenesis, accelerate the wound-healing process, and reduce inflammation. Therefore, SF has a wide range of applications in treating trauma, burns, and ulcers. In addition, SF is used in the development of drug delivery systems. Due to its good degradability and drug-modification properties, SF can be used as a carrier to control the release of drugs. This property makes SF promising, improving traditional drug-delivery systems, and developing novel drug delivery technologies. In conclusion, SF has a wide range of roles in biomedical applications. It can be used in tissue engineering, wound repair, and the development of drug delivery systems, providing valuable tools and materials for medical research and clinical practice. [26,27,28]. Therefore, SF-based biomaterials have been used as potential bio-polymer applications in bone/cartilage repair in tissue engineering (Figure 1) [29,30,31]. This paper introduces the biological characteristics of SF, its role in bone/cartilage injury, and its clinical applications.

## 2. Biological Properties of SF

Silk is currently an SF raw material that can be mass-produced and has been applied in clinical practice. As a natural fiber material, it mainly comprises two proteins: SF (SF) and silk sericin (SS). SF accounts for about 75%, and silk sericin accounts for about 25%. Silk has good biocompatibility and biodegradability [35,36]. To obtain pure SF, silk needs to undergo processes such as degumming, washing, and drying [37]. SF is a large molecular protein composed of 18 amino acids, of which glycine, alanine, and serine account for more than 80% of the total amino acid content [38,39]. It is widely distributed in the extracellular matrix and has various biological functions (Figure 2) [37,40].

### 2.1. Structure of SF

The molecular structure of SF has certain characteristics, consisting of a heavy (H) chain of 390 kDa and a light (L) chain of 26 kDa connected by disulfide bonds, as well as a glycoprotein (P25/30 kDa) secreted into the posterior silk gland [41,42,43]. The H chain accounts for most of the SF, with an amino acid composition of Gly (46%), Ala (30%), Ser (12%), Tyr (5.3%), and Val (1.8%) [44]. Another 4 kDa peptide encoded by the P25 gene is mainly associated with the H-L complex via hydrophobic interactions [45]. The genes encoding the three peptides are located on different chromosomes but appear to be coordinately regulated in the posterior silk gland [46,47]. In addition, interactions between the H chain and the L chain or P25 are crucial for the secretion of SF [48,49]. The main body of the SF chain consists of alternating crystalline and non-crystalline regions [50,51]. The crystalline region is dominated by the GAGAG sequence (Gly-Ala-Gly-Ala-Ala-Gly-Ser) with short side chains. The arrangement of amino acid residues in the amorphous region is complex and contains many amino acid residues with long side chains, such as tyrosine, lysine, and arginine. These residues are relatively hydrophilic, obstructing the regular assembly and crystallization of the chain segment, resulting in an irregularly coiled molecular conformation. The mechanical properties of SF can be regulated by changing the size, number, orientation, and arrangement of the crystalline (silk) and amorphous regions [52,53]. The primary crystal structures of SF are Silk I and Silk II, and the water-soluble and unstable Silk I can be transformed into Silk II with a β-fold structure that is insoluble in water under certain conditions [54,55].

### 2.2. Properties of SF

The stability and degradation of SF depend on factors such as temperature, pH, oxidation, enzyme action, light, and humidity. Reasonable control of these factors can prolong the stability of SF. The biological properties of SF are closely related to its structure [53,56]. Research has shown that SF can bind to receptors on cell membranes, thereby regulating biological processes such as cell proliferation, differentiation, and migration [40]. In addition, SF can also regulate the synthesis and degradation of the extracellular matrix, promoting the reconstruction and repair of the extracellular matrix [57,58]. In bone and cartilage tissues, SF is an important component that can regulate biological processes such as cell proliferation, differentiation, and migration [59]. The degradation properties of biomaterials directly affect the speed and quality of cartilage/osteochondral repair. As a kind of protein material, the degradation rate of SF-based biomaterials is mainly affected by proteases. Most proteolytic enzymes tend to degrade non-crystalline SF. This suggests that SF-based biomaterials with controlled degradability can be prepared by changing the content of crystalline structures [31]. The stability and degradation of SF depend on factors such as temperature, pH, oxidation, enzyme action, light, and humidity. Reasonable control of these factors can prolong the stability of SF. Due to its biocompatibility, adjustable degradation, unique biomedical and mechanical properties, ease of processing, and abundant supply, SF can be processed into gels [60], films [61], nanofibers [62], nanoparticles [63], and other materials that can be widely applied in drug delivery [64], tissue repair [65], and other fields.

### 2.3. Preparation Method of SF-Based Biomaterials

At present, the preparation methods of SF-based biomaterials mainly include 3D bioprinting, electrospinning, and freeze-drying. The main types of 3D bioprinting equipment include inkjet printing, extrusion printing, and laser-assisted printing [66,67]. Bio-inks are very important in 3D printing [68]. In general, when using 3D bioprinting to prepare SF-based biomaterials, it is necessary to modify SF bio-inks to enhance the biological activity of the SF, and the mechanical strength of SF-based biomaterials can be overcome by modifying SF-based bio-inks to meet the needs of cartilage/osteochondral repair [69,70]. Electrospinning can mix multiple matrices and combine the properties of the matrix to suit the needs of cartilage tissue engineering [71]. In addition, electrospinning can maintain the elasticity of SF, which is essential for cartilage/osteochondral cartilage tissue engineering [72]. In the process of preparing biomaterials via freeze-drying, technology freezes the solvent and then sublimates, with little effect on the solute, so freeze-drying also facilitates the carrying of drugs and growth factors for SF-based biomaterials [73].

### 2.4. The Main Types of SF-Based Biomaterials

The main types of SF-based biomaterials are hydrogels, scaffolds, and microcarriers. The native EMC-like microenvironment of hydrogels is suitable for loading cells to promote cartilage/osteochondral repair [74]. At the same time, the mechanical properties, shape, and swelling properties of hydrogels will change with the changes in temperature, pH value, and ion concentration, which can achieve the intelligent release of chondrogenic and osteogenic drugs and closely fit the interface of cartilage defects to improve the integration effect [75]. Scaffolds promote cartilage/osteochondral repair and regeneration by providing a specific microenvironment [76]. Compared with hydrogels, scaffolds have a fixed shape and higher mechanical strength, which can be used as an adjunct to cell therapy to promote cell attachment, growth, and differentiation [77]. Microcarriers generally refer to small spherical scaffolds suitable for cell culture, growth, and transport, as they not only promote cell growth and maintain the cell differentiation phenotype but also enable tissue regeneration through direct injection into the target site, enabling microcarriers to accelerate cartilage/osteochondral cartilage repair [78]. At the same time, microcarriers can be loaded with growth factors to promote cartilage/osteoblast adhesion and growth [79].

## 3. Osteoarthritic Articular Cartilage Model

Articular cartilage plays an important role in joint movement and is very finely and scientifically structured to suit different functional needs [80]. Articular cartilage mainly comprises a hyaline cartilage layer and a calcified cartilage layer [81]. In addition, the subchondral bone is located directly beneath the articular cartilage, and the hyaline cartilage, calcified cartilage, and subchondral bone together form the cartilage complex [82]. Hyaline cartilage is composed mainly of oval chondrocytes, an extracellular matrix (containing large amounts of proteoglycan), and type III collagen [83]. Articular cartilage is not innervated or vascularized, and its nutrients must be obtained from the joint fluid [84]. The core proteins that aggregate proteoglycans have covalently bound and strongly negatively charged glycosaminoglycan side chains that interact with the surrounding fluid environment through collagen and proteoglycan non-covalent linkages, giving articular cartilage its unique biomechanical properties [85]. The cells in the calcified cartilage zone are hypertrophic chondrocytes [86], and, because calcified cartilage is more dense and mineralized, its modulus of elasticity is at the megapascal level, between the kilopascal level of hyaline cartilage and the quarter-pascal level of bone tissue, which is equivalent to being a mechanical transition zone that prevents cartilage tissue from being damaged when subjected to excessive loading, and also effectively disperses the concentrated stress on cartilage tissue under shear [87]. In addition, the presence of a physiological demarcation line (tidal line) between the hyaline cartilage layer and the mineralized edge of the calcified layer indicates that the articular cartilage has significantly developed and that the growth-plate cartilage is fixed to the epiphysis, sometimes through a thin layer of calcified cartilage and tidal markings, while the hypertrophic edge does not form tidal markings and undergoes continuous vascular infiltration and endochondral ossification (EO) until the bone matures, and the growth plate is completely resorbed and replaced by bone [88].

The pathological changes of OA initially occur in the hyaline cartilage layer, which in turn causes changes in the calcified cartilage layer [89,90]. The onset of OA initially causes alterations in the spatial structure of the proteoglycan and collagen fibers of the hyaline cartilage layer, with swelling of the collagen fibers and an increase in free water [91,92]. The calcified cartilage layer, an important structure for conducting stress with the tide line under repeated compression, undergoes structural pathological changes, especially in the weight-bearing area, where microfractures appear [93]. The persistent production of microfractures causes the invasion of neovascular tissue and mineralization of the tideline, exacerbating the degenerative process of the hyaline cartilage [94,95]. As OA progresses, the loss of proteoglycans (PGs) from the cartilage tissue intensifies, and the collagen fibers continue to swell, causing the calcified layer to thin, micro-fissures to increase, and new capillaries to invade the calcified layer [96,97]. When cartilage tissue loses the protection of the calcified layer, a vicious cycle is formed, and the OA process is accelerated (Figure 3) [98].

## 4. The Role of SF in Bone/Cartilage Damage

SF plays a key role in chondrocyte differentiation. SF is a protein found primarily in collagen fibers that helps form and maintain the structure and function of cartilage tissue. During chondrocyte differentiation, SF forms a complete extracellular matrix environment by interacting with other extracellular matrix components such as collagen and glycosaminoglycans. This environment provides the necessary support and structure for chondrocytes to grow and differentiate. In addition, SF is also involved in key pathways that regulate cell signaling and cell function. It can affect cell proliferation, differentiation, and migration by binding to cell surface receptors (Figure 4). SF can also interact with intracellular signaling pathways, such as TGF-β and BMP, to regulate the direction and speed of cell differentiation [99,100]. Recent studies have shown that SF plays an important role in the repair and regeneration of bone/chondrogenic tissue (Table 1) [101,102].

### 4.1. The Role of SF in Bone Tissue

Bone tissue is one of the hardest tissues in the body, and damage and lesions often lead to fractures, osteoporosis, and other diseases [103,104]. Recent studies have shown that SFs play an important role in bone tissue [105,106,107,108]. As a natural polymer material with good elasticity, tensile strength, biocompatibility, and biodegradability, SF can provide sufficient space for the growth and differentiation of bone cells, thus promoting the generation of new bone tissue [109,110,111,112,113]. SF has been extensively studied in bone TE because of its high toughness, mechanical strength, and proven biocompatibility. Meinel et al. combined bone tissue engineering, gene therapy based on human mesenchymal stem cells (MSCs), and SF biomaterials to investigate the effect of viral transfection on MSC osteogenic properties in vitro, and showed that RSF scaffolds promote osteogenic differentiation of human mesenchymal stem cells (HMSCs) in vitro [113]. Meanwhile, Meinel et al. implanted porous SF-based scaffolds into cranial defects in mice, demonstrating bone tissue regeneration using silk-based implants with engineered bone, and expanding the selection of protein-based bone implant materials through mechanical stability and durability [114]. RSF scaffolds can be combined with other biomaterials, such as collagen or calcium phosphate-based inorganic components, to enhance osteogenic properties [115]. To increase the success of bone regeneration, accelerated angiogenesis is required [116]. Farokhi et al. used bio-mixed SF/calcium phosphate/PLGA nanocomposite scaffolds as a vascular endothelial growth factor (VEGF) to explore the efficacy of the delivery system, which also showed good effects [117].

Bioactive molecules and active cells are a hot topic in tissue engineering research, with various molecules and cells providing additional regulatory cues to guide cell differentiation and functional bone regeneration [118]. Growth factors, drugs, and different stem cells have been introduced into filament-based scaffolds to promote bone formation (Figure 5) [119].

#### 4.1.1. Bioactive Factor-Based Biomaterials

Bioactive factor-based biomaterials can release bioactive factors and promote bone tissue repair by regulating cell proliferation and differentiation [120]. Drug-loaded beads and colloidal crystals can be used to achieve a controlled release of bioactive factors through microstructural alterations to improve bone repair [121,122]. Shen et al. developed an SF/nano-hydroxyapatite (nHAp)-based scaffold with sequential and sustained release of SDF-1 and BMP-2 in SF/nHAp scaffolds with synergistic effects on bone regeneration [123]. Exosome-encapsulated silk frames have also been shown to promote the recovery of bone defects in vivo [124].

#### 4.1.2. Biodegradable Polymer-Based Biomaterials

Biodegradable polymeric biomaterials are biomaterials with good biocompatibility and tunability, which can be modified in terms of composition, structure, and physicochemical properties to achieve a variety of different functions, such as cell adhesion, biodegradation, and drug retardation [125,126,127]. Currently, common biodegradable polymer-based biomaterials include polylactic acid and polycaprolactone [128,129]. Diaz-Gomez et al. prepared composite scaffolds using various combinations of PCL, SF, and nanohydroxyapatite (nHA), confirming the synergistic effect of silk and nHA on the extent of bone repair [130]. As a biodegradable polymer film, the silk in protein/chitosan composite film can be used not only as a metal implant coating for bone injury repair but also as a tissue engineering scaffold for skin, cornea, adipose, and other soft tissue injury repair, while the silk in protein/chitosan film not only provided a comparable environment for the growth and proliferation of rat bone marrow-derived mesenchymal stem cells but also promoted their osteogenic and lipogenic differentiation [131]. Biodegradable polymeric composites of SF are widely used in tracheal tissue engineering [132], wound dressings [133], and materials for biomedical applications [134].

#### 4.1.3. Calcium- and Phosphorus-Based Biomaterials

Calcium and phosphorus biomaterials are biomaterials that can form, in vivo, similar to bone tissue, and can promote bone tissue repair, for example, by promoting the proliferation and differentiation of bone cells [135]. For example, hydroxyapatite and calcium tri-calcium phosphate can be used as carriers of bone growth factors to promote bone tissue repair through the slow release of bone growth factors [136,137]. At the same time, the significant anti-stress properties of SF in concert with β-tricalcium phosphate provided a good bionic environment for bone marrow MSCs [138]. The SF/calcium phosphate material has good biocompatibility and mechanical properties as a bone repair scaffold, providing a good growth space for osteoblast differentiation. In addition, the injectability exhibited by the hydrogel and its use as a drug carrier allows the hydrogel to fit closely to the interface of the cartilage defect, thus improving the integration effect [139].

### 4.2. The Role of SF in Cartilage Tissue

Cartilage tissue is an elastic connective tissue whose main role is to cushion and support bone [140,141]. Injuries and lesions to cartilage tissue often lead to diseases such as OA and cartilage lesions [142]. SF is associated with the extracellular matrix widely distributed in cartilage tissue, and its main role is to maintain the structure and function of the cartilage tissue [143]. SFs can promote the proliferation and differentiation of chondrocytes and the generation of new cartilage tissue [144]. In addition, SFs can regulate the apoptosis and survival of chondrocytes, and promote the repair and regeneration of cartilage tissue [145].

#### 4.2.1. Application of SFs in Cartilage Tissue Engineering

Cartilage tissue engineering is a method of repairing and reconstructing cartilage tissue using biomaterials such as cells, matrix materials, and growth factors [146]. SF is a natural protein that has many similarities with the protein structure of human skin and has good biocompatibility and biodegradability [28]. In cartilage tissue engineering, SF can be used to construct biocompatible, biodegradable scaffold materials, and can likewise be used as surface modifiers to enhance scaffold surface cell adhesion and proliferation properties (Figure 6) [147]. Studies have shown that SFs can promote the proliferation and differentiation of chondrocytes, thereby promoting the growth and repair of cartilage tissue [101]. In addition, SFs can improve the strength and stability of cartilage tissue, thus increasing the success rate of cartilage tissue engineering [148]. Additionally, silk in proteins has good water retention and can provide a good environment under growth conditions to aid chondrocytes in nascent tissue growth and repair [149]. Saha et al. evaluated the role of mulberry and non-mulberry laminar filament biomaterials in cartilage or bone induction using human bone marrow stromal cells (hBMSC) in vivo and in vitro, showing good bone induction [150]. In cartilage tissue engineering, combining SF with other biomaterials, such as gelatin and alginate, supports the fabrication of scaffold materials with good biocompatibility and biodegradable properties that can provide skeletal support for the generation of new tissue, and can degrade into harmless metabolites with reduced adverse effects on the human body [151,152]. Li et al. developed a silk-composite hydrogel of SF and carboxymethyl chitosan (CMCS), which supported the adhesion, proliferation, glycosaminoglycan synthesis, and chondrogenic phenotype of rabbit articular chondrocytes, and the subcutaneous implantation of the hydrogel in mice showed no infection or local inflammatory response, indicating good biocompatibility in vivo [153]. Liu et al. used electrostatic spinning to prepare a fibrillar SF/poly L-lactic acid (PLLA) scaffold, showing good adhesion, biocompatibility, and cytocompatibility [154]. Overall, SF, as an important matrix material, has good prospects for application in cartilage tissue engineering. With a better understanding of silk, we will be better able to apply this natural protein to promote cartilage tissue repair and regeneration.

#### 4.2.2. SF in the Treatment of Patients with OA

OA is a serious disease derived from the degeneration of cartilage tissue and, if necessary, requires surgical intervention. Tissue engineering using stem cell graft scaffolds is an attractive approach and a challenge for orthopedic surgery [137,138]. SFs can also reduce pain and inflammatory responses in patients with arthritis, thereby relieving the symptoms of arthritis. Wang et al. prepared silk/BDDE hydrogel balls using oil/water (o/w) emulsification and evaluated their biocompatibility and biodegradability in vivo. The silk/BDDE hydrogel ball was demonstrated to be biocompatible and can be used as a lubricant for the treatment of OA, as well as for pain relief and the sustained release of drugs for future OA treatment [155]. In clinical trials, Sharafat-Vaziri et al. evaluated an autologous chondrocyte and collagen/silk heart protein scaffold consisting of newly engineered tissues to repair osteochondral defects, showing great coverage and integration of the grafts in patients without effusion, edema, and reduced cartilage formation signals [156]. Jaipaew et al. prepared SF/hyaluronic acid (HA) scaffolds, with different SF/HA (*w*/*w*) ratios via freeze-drying, which were suitable for OA surgery [157]. Thus, SF-based biomaterials have great promise in OA surgery.

#### 4.2.3. Application of SFs in Drug Delivery

SF is a biomaterial that has been extensively studied in tissue engineering and drug delivery (Figure 7). As a carrier material, SF has high permeability, and its microporous structure can promote the penetration of drugs and improve the delivery efficiency of drugs. At the same time, SF can be degraded and absorbed by the human body, avoiding the risk of secondary surgery. Additionally, SF can protect the drug from degradation and inactivation during delivery and improve the stability of the drug. Moreover, SF has low immunogenicity and histocompatibility, reducing rejection of the drug delivery system [158]. At present, SF can be used as a carrier for drugs in OA drug delivery, enveloping drugs in nanoparticles or microspheres to increase the efficiency and stability of drug delivery. Additionally, it can be combined with growth factors or cytokines through controlled release, promote the proliferation and differentiation of chondrocytes, and promote the repair and regeneration of joint cartilage. At the same time, it can combine with stem cells or osteoblasts to form a tissue-engineered scaffold for the repair and regeneration of joint cartilage [159,160,161]. In addition, SFs can be used in combination with other biomaterials to further enhance their drug delivery [162]. Ratanavaraporn loaded previously developed gelatin/SF microspheres with curcumin, and the gelatin/SF microspheres encapsulated with curcumin delayed cell destruction in joint and synovial tissue, which showed prolonged anti-inflammatory effects compared to rapidly degrading gelatin microspheres [163]. Red-modified silk nanoparticles were fabricated by Sharma et al. and loaded with gentamicin as a deposition material on titanium surfaces, which showed better killing of S. aureus on the titanium surfaces [164]. Hassani Besheli et al. constructed a sustained drug delivery system by loading vancomycin (VANCO) in silk nanoparticles to treat severe osteomyelitis [165]. Thus, in OA drug delivery, SF can play an important role (Table 2).

### 4.3. Prospects for SF in Clinical Applications

As a natural polymer with good biocompatibility, SF has been widely used in biomedical applications, including ophthalmic surgical sutures, artificial corneas, artificial tendons, orthopedic ligaments, cartilage engineering, artificial skin for wound surfaces in traumatology, and anticoagulation stents in cardiology [174,175,176]. Although there is extensive research into the application of SFs in different base materials, most of it is still at the laboratory research stage, and few products have been successfully commercialized and are actually used in clinical treatment (Table 3). SERI surgical scaffolds, Silk Voice injections, and silk-substituted isoserine trauma dressings are the products in routine clinical use today (Table 4).

With increasing in-depth research on SF, its application in the treatment and repair of bone and cartilage tissues is becoming more promising. In regenerative medicine, the binding of SF to cells or growth factors can promote cell attachment, proliferation, and differentiation, while providing structural support to help rebuild tissues. The biocompatibility and biodegradability of SF fibers make them ideal materials for manufacturing artificial blood vessels, and SF can be used in the preparation of biomedical dressings to promote the healing process of wounds. SF dressings provide a protective physical barrier to promote cell regrowth and tissue repair, and SF can be used as a vehicle for drug delivery systems. SF has good drug adsorption and sustained release performance, which can control the release rate and dose of drugs and improve the efficacy and bioavailability of drugs; SF can also be used to prepare artificial bone substitutes for bone-tissue engineering and bone repair. Moreover, an SF scaffold can promote the attachment and growth of bone cells and accelerate the process of bone regeneration and repair. SF has wide application prospects in regenerative medicine, which can promote tissue regeneration and repair and provide new treatments and means for disease treatment and rehabilitation [155,177,178]. In terms of applications, most are concentrated in the fields of wound repair dressings [179,180] and orthopedic repair materials [31]; in terms of morphology, SF membranes [181] and SF scaffolds [182] are the materials with more applications. SF can be combined with other biomaterials to form composite scaffolds, mimic the natural in vivo environment, increase the in vivo fusion potential of scaffolds, implant destination cells to accelerate the healing and regeneration of trauma sites, etc. It can be modified into different scaffolds such as injectable and printable gels, porous sponges, and electrostatically spun two- and three-dimensional structures [130,134]. The advantages of SF therapy are its obvious effects, low side effects, and high safety, and more possibilities for SF-based scaffolds can be expected through new modification techniques. In the future, the application of SF in the treatment and repair of bone and cartilage tissues will be even more promising, while more research is needed to explore its mechanism of action and dose determination.

## 5. Conclusions and Outlook

Repairing cartilage and osteochondral damage due to OA has always been a challenge for clinicians. Currently, tissue engineering techniques have been the subject of much research and have great potential in bone/cartilage repair. SF, as a natural material, is widely used in tissue engineering due to its inexpensive availability, excellent biocompatibility, unique mechanical properties, and desirable processing properties. This paper summarizes the progress of research on SF biomaterials in the field of bone/cartilage repair. Compared with other types of biomaterials, SF has great research value and broad application prospects. SF-based biomaterials have more robust mechanical properties through loading peptides, gene-editing means, exosomes, nanoparticles, and growth factors, which are more conducive to cell adhesion and growth and enhance bone/cartilage repair. For example, Ding et al. significantly improved the cell recruitment ability of SF/HA scaffolds by loading bone morphogenetic protein-2 (BMP-2) onto SF/HA scaffolds to accelerate osteochondral repair [183], proliferation, and differentiation, promoting bone repair and cartilage repair [184].

SF has good biocompatibility, biodegradability, and bio-absorbability, and has been widely used in the medical field in recent years. Currently, clinical applications regarding SF biomaterials are mainly focused on wound healing and wound dressings [114]. In addition, there have also been clinical studies on SF scaffolds for meniscal cartilage repair, with some success in recent years [185]. However, for OA and cartilage regeneration, SF has some limitations. First of all, OA is a joint disease whose main features are cartilage degeneration and joint inflammation. While SF can help maintain cartilage structure, it does not prevent or reverse cartilage degeneration. This means that SF has a limited therapeutic effect on OA. Secondly, SF also has certain limitations on cartilage regeneration. Cartilage regeneration is a complex biological process that includes steps such as stem cell differentiation, cell proliferation, and secretion of a matrix. Although SF has a certain effect on cartilage structure, it does not directly promote cartilage regeneration. When treating cartilage damage or joint degenerative diseases, stem cell therapy, biomaterial implantation, or surgery are often required to promote cartilage regeneration and repair. Overall, while SF plays an important role in maintaining cartilage structure and function, it has certain limitations in OA and cartilage regeneration. The treatment of these conditions often requires a combination of factors and treatments to achieve the best results [186].

In conclusion, SF-based biomaterials have broad application potential in the treatment of OA, and future research can combine SF biomaterials with other bioactive molecules, such as growth factors and gene therapy, to improve the effect of cartilage regeneration. At the same time, according to the specific situation of patients, the personalized design of SF biomaterials can help achieve better treatment results. More importantly, as the preparation process of SF biomaterials is further improved, the mechanical properties and stability of the materials are improved, and their degradation rate in vivo is prolonged. Although SF biomaterials have achieved some success in the treatment of OA, they still face some challenges and limitations. More clinical research and scientific exploration will help further develop and refine this treatment, providing better treatment options for patients with OA.

## Figures and Tables

**Figure 1 biomedicines-11-02244-f001:**
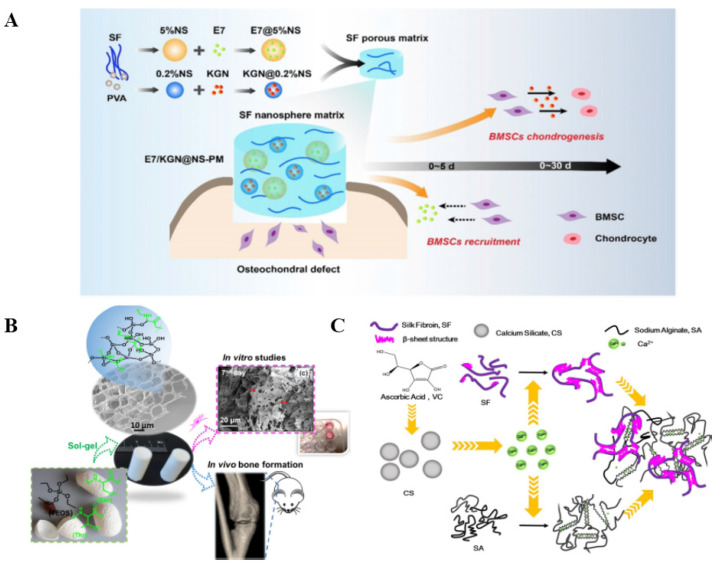
SF-based biomaterials in cartilage/osteochondral repair. (**A**) Sequential release of E7/KGN from silk nanosphere matrix in osteochondral defect repair. Reprinted/adapted with permission from Ref. [32], copyright 2020, Elsevier. (**B**) Mechanically strong silica-silk bioaerogel for bone regeneration. Reprinted/adapted with permission from Ref. [33], copyright 2019, American Chemical Society. (**C**) Fabrication of silk/calcium silicate/sodium alginate composite scaffolds. Reprinted/adapted with permission from Ref. [34], copyright 2018, Elsevier.

**Figure 2 biomedicines-11-02244-f002:**
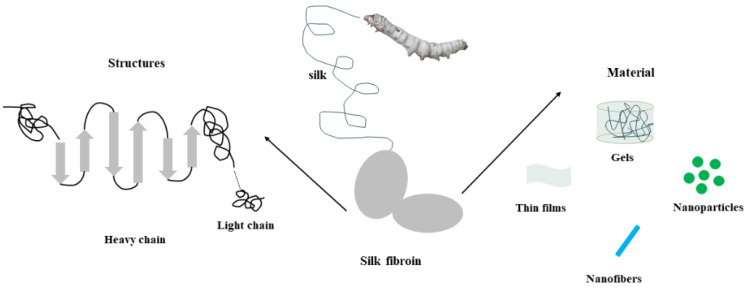
SF structure and SF-based materials.

**Figure 3 biomedicines-11-02244-f003:**
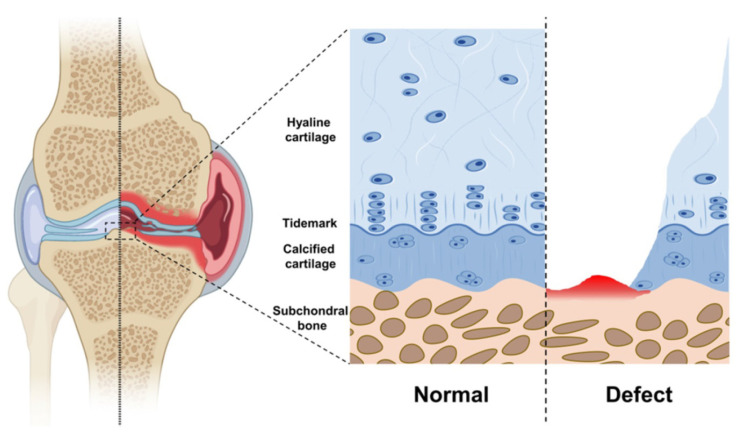
Schematic diagram of articular cartilage. Reprinted/adapted with permission from Ref. [31], copyright 2020, Ivyspring.

**Figure 4 biomedicines-11-02244-f004:**
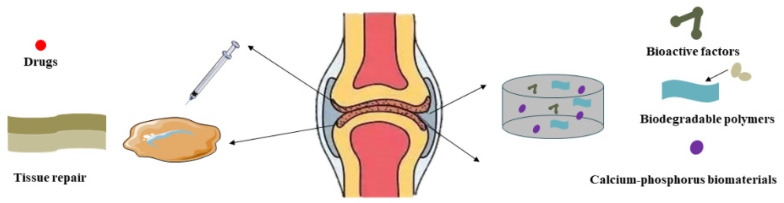
The role of SF in bone/cartilage damage.

**Figure 5 biomedicines-11-02244-f005:**
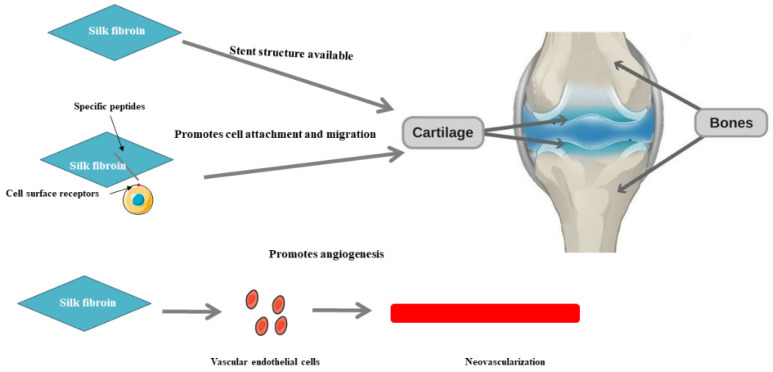
The role of silk fibroin in cartilage tissue engineering.

**Figure 6 biomedicines-11-02244-f006:**
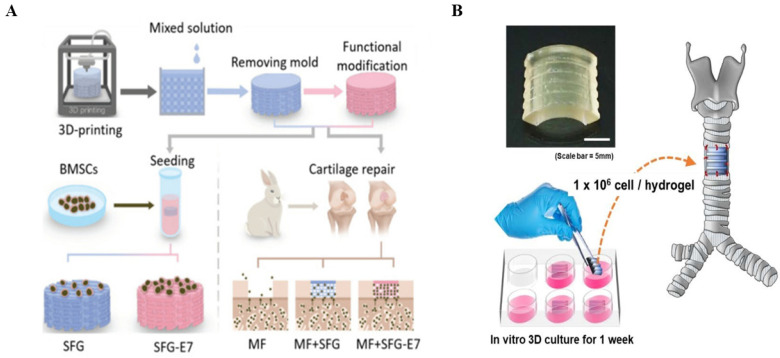
In vivo implantation of SF-based biomaterials. (**A**) Schematic diagram illustrating the 3D fabrication of a scaffold made via bioprinting for in vivo implantation. Reprinted/adapted with permission from Ref. [151], copyright 2017, John Wiley and Sons. (**B**) Silk-GMA hydrogel transplantation loaded with chondrocytes. Reprinted/adapted with permission from Ref. [140], copyright 2020, Elsevier.

**Figure 7 biomedicines-11-02244-f007:**
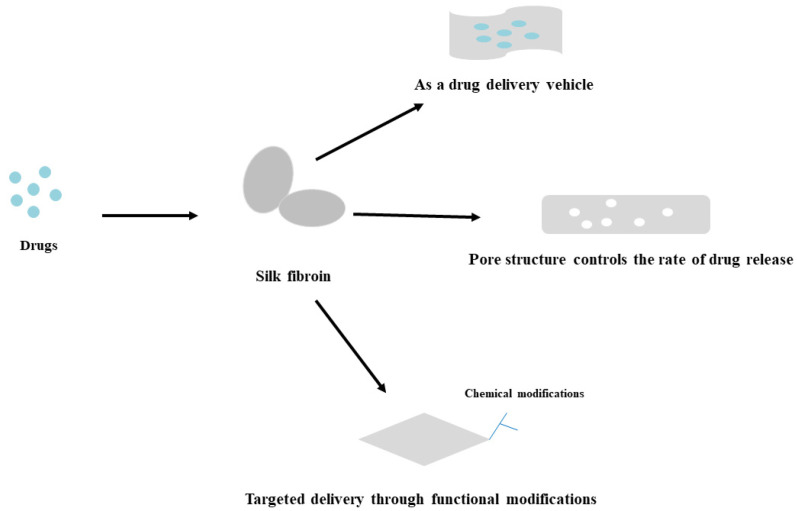
The role of SF in drug delivery.

**Table 1 biomedicines-11-02244-t001:** Application of SF in bone tissue regeneration.

Application Areas	Mechanism of Action	Application Results	References
Bone regeneration and repair	Promotes osteoblast proliferation and differentiation, bone matrix production, and epiphyseal migration	Promotes the speed of fracture healing, enhances fracture stability, and promotes bone defect repair	[31,69]
Cartilage regeneration and repair	Promotes proliferation and differentiation of chondroblasts and synthesis of collagen and cartilage matrix	Promotes healing of cartilage defects and improves cartilage tissue structure and functional recovery	[99,100]
Bone implant repair	Provides an extracellular matrix scaffold to improve the biocompatibility and adhesion of bone implants	Enhances the bonding of the bone implant to the surrounding tissue and promotes stability and growth of the bone implant	[101]
Oral periodontal restoration	Promotes the growth of dental bone attachment tissue and soft tissue repair	Improves the effectiveness of periodontitis treatment and promotes oral wound healing	[102]
Other applications	Various tissue engineering repairs, angiogenesis, immunomodulation, etc.	SF has potential for a wide range of applications in tissue engineering and regenerative medicine	[44]

**Table 2 biomedicines-11-02244-t002:** Application of SF in drug delivery.

Application Areas	SF in Drug Delivery	References
Oncology treatment	To deliver drugs to tumor tissue, SFs are used as carriers to improve the stability and bioavailability of drugs	[166]
Wound healing	SFs promote cell migration, proliferation, and repair during wound healing and can be used to prepare drug-delivery systems to promote wound healing	[167]
Treatment of blood disorders	SFs provide reliable carriers for the transport and release of drugs for the treatment of blood disorders, such as anticoagulants and anti-platelet agents	[168]
Treatment of neurological disorders	SFs can be used to deliver drugs for the treatment of neurological disorders, such as neuroprotective agents and anti-epileptic drugs, to promote the protection and repair of nerve cells	[169]
Skin beauty and treatment	SFs are widely used in cosmetic skin products and delivery systems for therapeutic drugs to improve skin texture, promote wound healing, reduce scar formation, etc.	[170]
Infectious disease control	SFs are used as carriers for drug delivery systems to deliver antiviral, antibacterial, and antifungal drugs, improving their efficacy and bioaccessibility	[171]
Treatment of orthopedic diseases	SFs are used to deliver drugs for treating orthopedic diseases, such as bone growth factors and anti-inflammatory drugs, and to promote the growth and repair of bone cells	[106,107]
Cardiovascular disease treatment	SFs are used as carriers in drug delivery systems for the delivery of drugs treating cardiovascular disease, such as anti-hypertensives and anti-heart failure drugs, to alleviate the symptoms of cardiovascular disease	[172]
Immune disease treatment	SFs can be used in drug delivery systems to deliver drugs for treating immune diseases, such as anti-inflammatory drugs and immunomodulators, to regulate the function of the immune system and treat diseases	[102]
Dental treatment	SFs are widely used in dental therapeutic drug delivery systems for the delivery of antibacterial drugs, natural anti-inflammatory agents, and bone growth factors to promote dental restoration and healing	[173]

**Table 3 biomedicines-11-02244-t003:** Selected SF material development companies.

Company Name	Region	SF Products
Sofregen Inc.	US	SERI surgical stents
		Injectable fillers
Vaxess Technologies Inc.	US	Drug and vaccine delivery
Evolved by nature	US	Skincare products, textile coatings, topical ophthalmic treatments, etc.
Cocoon Biotech Inc.	US	Drug delivery systems (hydrogels, osteoarticular microspheres, etc.)
Kraig Biocraft Laboratories Inc.	US	Special textiles
Oxford Biomaterials Ltd.	UK	Artificial blood vessels
Orthox Ltd.	UK	Meniscal repair stents, tissue stents
Suzhou Semtex Biotechnology Co.	China	Injectable gels, stents, and dressings
Suzhou Suhao Biomaterials Technology Co.	China	Trauma dressings
Zhejiang Xingyue Biotechnology Co.	China	Raw materials such as SF gels, microspheres, solutions, and sponges
		SF skin rejuvenation mask

**Table 4 biomedicines-11-02244-t004:** Some approved SF products.

Product Type	Trademarks	Uses	Time to Market
SERI Surgical Stent	Allergan	Full body contouring, brachioplasty, abdominoplasty, breast fixation, breast reconstruction, etc.	2013
Silk Voice Injection	Sofregen	Vocal cord dielectricity and vocal cord insufficiency	United States, 2018
Silk-substituted isoserine wound dressing	Soho Biotechnology Co.	Wound healing	China, 2012

## Data Availability

The data presented in this study are available on request from the corresponding author.

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
