# Peer review of "Evaluation and Application of Silk Fibroin Based Biomaterials to Promote Cartilage Regeneration in Osteoarthritis Therapy"

_biomedicines, 2023, doi:10.3390/biomedicines11082244_

Round 1
Reviewer 1 Report
Title: “Evaluation and Application of Silk Fibroin Based Biomaterials 2 toPromote Cartilage Regeneration in Osteoarthritis Therapy” Authors: Authors: Xudong Su, Li Wei, Zhenghao Xu, Leilei Qin, Jianye Yang,
Yinshuang Zou, Chen Zhao, Li Chen, Ning Hu Comments to the Author:
This review presents a comprehensive summary of the biological features of silk fibroin, the role it plays in bone and cartilage lesions, and its outlook for clinical uses to offer new perspectives and references for the field of bone and cartilage restoration. However, the article contains several criticisms that raise questions about the accuracy and completeness of the discussion.
Several points are listed below:
1: The title is confusing and does not say what the authors actually want to say. It should be fundamentally rewritten!
2: The abstract should be rewritten as it contains many repetitions and appears Unstructured.
3: A clear graphical abstract should be created that is meaningful in terms of content.
4: The introduction should be restructured from scratch.
5: The novelty of the article should be clearly emphasized, and the reasons for focusing on obesity and microbiota, diabetes mellitus, and osteoarthritis can be explained.
6: Limitations related to Silk Fibroin for Osteoarthritis and cartilage regeneration should be discussed in more detail.
7: For the biomedical application of silk protein, it is suggested to describe the role of silk protein in detail.
8: It is recommended that “Structure, components and properties of silk” section be divided into three paragraphs or add subtitles to make information in the passage clearer.
9: It is recommended that the authors add tables to summarize the applications of silkworm proteins in the sections of “Organ and tissue regeneration” and “Drug delivery systems and regenerative medicine”.
10: In the final part, it is recommended to summarize the biomedical applications of several types of silk proteins in regenerative medicine.
11: Discussion: I miss a detailed discussion of the entire work. I suggest that you adjust the content of the discussion to the results.
12: The paper contains many complicated abbreviations. A list of the most
important abbreviations would be helpful for the reader.
13: English revision is needed
Moderate editing of English language required
Author Response
Dear Reviewer:
Thank you very much for reviewing our manuscript “Silk Fibroin Based Biomaterials for Cartilage Regeneration in Osteoarthritis Therapy” (Manuscript ID: biomedicines-2465044). It is our pleasure to have the opportunity to publish papers in Biomedicines. Besides, Thank you very much for your kind letter and encouragement again, along with the comments from the reviewer and you concerning our manuscript “Silk Fibroin Based Biomaterials for Cartilage Regeneration in Osteoarthritis Therapy” (Manuscript ID: biomedicines-2465044). We have thoroughly considered all the comments and substantially revised our manuscript. The point-to-point answers and explanations for all comments were listed in separate pages following this letter.
Once again thank you for your kind consideration, and your efforts on this manuscript are greatly appreciated.
Best wishes to you!
Yours sincerely,
Prof. Li Chen and Ning Hu.
The detailed response is appended below.
Comment 1: The title is confusing and does not say what the authors actually want to say. It should be fundamentally rewritten!
Reply: Let me sincerely thank you for your professional review and your suggestions and recognition of this research work. Based on your suggestion, I have made a revision to the title and hope to get your approval (Please see: lines 2-3).
Comment 2: The abstract should be rewritten as it contains many repetitions and appears Unstructured.
Reply: Thank you for your professional advice, we read the summary carefully and have redescribed it (Please see: lines 24-34).
Comment 3: A clear graphical abstract should be created that is meaningful in terms of content.
Reply: Indeed, as you said, a graphical summary will make the article clearer. Based on your suggestions, we have inserted them in the appropriate places (Please see: line 37).
Comment 4: The introduction should be restructured from scratch.
Reply: Thank you very much for your suggestion of my manuscript, I have rearranged the introduction (Please see: lines 41-94).
Comment 5: The novelty of the article should be clearly emphasized, and the reasons for focusing on obesity and microbiota, diabetes mellitus, and osteoarthritis can be explained.
Reply: Thank you very much for your professional advice on my article. Our manuscript is dedicated to summarizing the study of silk fibroin biomaterials in osteoarthritis, which is closely related to obesity and diabetes. Our manuscript focuses on silk fibroin in bone and joint, so it may not be a careful description of the relevant indicators you propose. Your advice is very professional.
Comment 6: Limitations related to Silk Fibroin for Osteoarthritis and cartilage regeneration should be discussed in more detail.
Reply: Thank you very much for your suggestion of my manuscript, I've added to the content (Please see: lines 477-488).
Comment 7: For the biomedical application of silk protein, it is suggested to describe the role of silk protein in detail.
Reply: Thank you very much for your suggestion of my manuscript, Regarding the use of silk fibroin in biomedicine, I have supplemented it (Please see: lines 432-445).
Comment 8: It is recommended that “Structure, components and properties of silk” section be divided into three paragraphs or add subtitles to make information in the passage clearer.
Reply: Thank you very much for your suggestion of my manuscript, your suggestions can make the manuscript clearer, so I have revised it (Please see: lines 108-130).
Comment 9: It is recommended that the authors add tables to summarize the applications of silkworm proteins in the sections of “Organ and tissue regeneration” and “Drug delivery systems and regenerative medicine”.
Reply: Your suggestions have fleshed out my article, I have made the addition of the table, thank you very much for your professional advice (Please see: lines 207-208, 410-411).
Comment 10: In the final part, it is recommended to summarize the biomedical applications of several types of silk proteins in regenerative medicine.
Reply: Based on your suggestions, I have supplemented the content of silk fibroin in regenerative medicine, thank you very much for your suggestions (Please see: lines 432-445).
Comment 11: Discussion: I miss a detailed discussion of the entire work. I suggest that you adjust the content of the discussion to the results.
Reply: Once again, we thank you for your efforts on our manuscript. We've reworked the discussion section (Please see: lines 477-500).
Comment 12: The paper contains many complicated abbreviations. A list of the most important abbreviations would be helpful for the reader.
Reply: Your suggestion is very correct, and we have rearranged the full-text abbreviations.
Comment 13: English revision is needed.
Reply: We apologize for the deficiencies in our grammar, so we have re-polished the article and thank you again for your efforts on our manuscript.
Reviewer 2 Report
The maniscript is interesting, well illustrated and generally well written. Only minor comments are needed. In particular:
Lines 43-45: It deserves to be pointed out that cartilage and subchondral bone degeneration in Osteoarthritis is due to increased metalloproteases and inflammatory cytokines characterizing this pathology ( as described in PMID: 35131488 ). This is an important point to add since it can further highlight the interesting studies discussed by the authors.
A table resuming the studies discussed in each section would help to better understand the manuscript
An accurate revision of typing errors is recommended
Author Response
Dear Reviewer:
Thank you very much for reviewing our manuscript “Silk Fibroin Based Biomaterials for Cartilage Regeneration in Osteoarthritis Therapy” (Manuscript ID: biomedicines-2465044). It is our pleasure to have the opportunity to publish papers in Biomedicines. Besides, Thank you very much for your kind letter and encouragement again, along with the comments from the reviewer and you concerning our manuscript “Silk Fibroin Based Biomaterials for Cartilage Regeneration in Osteoarthritis Therapy” (Manuscript ID: biomedicines-2465044). We have thoroughly considered all the comments and substantially revised our manuscript. The point-to-point answers and explanations for all comments were listed in separate pages following this letter.
Once again thank you for your kind consideration, and your efforts on this manuscript are greatly appreciated.
Best wishes to you!
Yours sincerely,
Prof. Li Chen and Ning Hu.
The detailed response is appended below.
Comment 1: It deserves to be pointed out that cartilage and subchondral bone degeneration in Osteoarthritis is due to increased metalloproteases and inflammatory cytokines characterizing this pathology (as described in PMID: 35131488 ). This is an important point to add since it can further highlight the interesting studies discussed by the authors.
Reply: Let me sincerely thank you for your professional review and your suggestions and recognition of this research work. Based on your suggestion, I have made a revision to the title and hope to get your approval (Please see: lines 48-52).
Comment 2: A table resuming the studies discussed in each section would help to better understand the manuscript.
Reply: Thank you very much for your professional advice, your suggestions have fleshed out my article, I have made the addition of the table (Please see: lines 207-208, 410-411).
Comment 3: An accurate revision of typing errors is recommended.
Reply: We are very sorry for the clerical error due to our mistakes, we have carefully read and revised the full text, and thank you very much for your suggestions for my manuscript.
Reviewer 3 Report
This presented manuscript by Su and coauthors aims to describe the current knowledge about silk fibroin usage as a component of biomaterials dedicated to cartilage regeneration. The manuscript can be potentially interesting, however in my opinion, it requires some modifications.
First of all, this is the review and it should not contain the figures taken from other publications – my suggestion is: all figures need to be prepared as new by the Authors, therefore they need to be original. In the text there is no matching/correlation to included figures.
The topic is generally concentrated on silk protein, therefore I suggest to add information about the production methods of this biocomponent. The types of silk-based materials (including types of scaffolds) and their mechanical/physical/chemical features should be added. The Authors mentioned about biodegradation of silk-based materials -there need to be added information of degradation products and their activity, time of silk-based products stability, solutions for some problems depending on the type of silk-scaffolds used.
In case of biological features – there is almost no information presented on the molecular level – this aspect needs to be improved by more detailed mechanism description (i.e., what about differentiation or redifferentaiation of chondrocytes, types of receptors, cells migration properties, transcription factors, enzymes activity?), as well as the quantitative data should be added (i.e. concentration/ dose, time of incubation, types of cells/issues). Additionally, the information is not clearly presented as obtained from in vitro or in vivo experiments – it obligatory needs to be rewritten. Please, add and comment the results from clinical studies.
I strongly suggest to reread the manuscript carefully, since within the text very often are repeated general information (maybe because some parts were prepared by different coauthors) and information not related to cartilage tissue – it needs to be modified.
Please explain the sentence in lines 290-292, since silk fibroin is not a component of human extracellular matrix.
The drug -delivery chapter (4.2.3) seems to have more information about anti-cancer activity – please, reconsider this chapter modification.
In general, I suggest the major revision.
Author Response
Dear Reviewer:
Thank you very much for reviewing our manuscript “Silk Fibroin Based Biomaterials for Cartilage Regeneration in Osteoarthritis Therapy” (Manuscript ID: biomedicines-2465044). It is our pleasure to have the opportunity to publish papers in Biomedicines. Besides, Thank you very much for your kind letter and encouragement again, along with the comments from the reviewer and you concerning our manuscript “Silk Fibroin Based Biomaterials for Cartilage Regeneration in Osteoarthritis Therapy” (Manuscript ID: biomedicines-2465044). We have thoroughly considered all the comments and substantially revised our manuscript. The point-to-point answers and explanations for all comments were listed in separate pages following this letter.
Once again thank you for your kind consideration, and your efforts on this manuscript are greatly appreciated.
Best wishes to you!
Yours sincerely,
Prof. Li Chen and Ning Hu.
The detailed response is appended below.
Dear Reviewer:
Thank you very much for reviewing our manuscript “Silk Fibroin Based Biomaterials for Cartilage Regeneration in Osteoarthritis Therapy” (Manuscript ID: biomedicines-2465044). It is our pleasure to have the opportunity to publish papers in Biomedicines. Besides, Thank you very much for your kind letter and encouragement again, along with the comments from the reviewer and you concerning our manuscript “Silk Fibroin Based Biomaterials for Cartilage Regeneration in Osteoarthritis Therapy” (Manuscript ID: biomedicines-2465044). We have thoroughly considered all the comments and substantially revised our manuscript. The point-to-point answers and explanations for all comments were listed in separate pages following this letter.
Once again thank you for your kind consideration, and your efforts on this manuscript are greatly appreciated.
Best wishes to you!
Yours sincerely,
Prof. Li Chen and Ning Hu.
The detailed response is appended below.
Comment 1: First of all, this is the review and it should not contain the figures taken from other publications – my suggestion is: all figures need to be prepared as new by the Authors, therefore they need to be original. In the text there is no matching/correlation to included figures.
Reply: Thank you very much for your professional advice. In the process of writing the article, because the experimental data related to this manuscript has not yet been completed, there are no original pictures when writing the review, but the pictures that appear in the article have been copyrighted, and we hope to be recognized by you.The copyright of the picture has been obtained and is attached for your viewing. Thank you again for your professional advice.
Comment 2: The topic is generally concentrated on silk protein, therefore I suggest to add information about the production methods of this biocomponent. The types of silk-based materials (including types of scaffolds) and their mechanical/physical/chemical features should be added. The Authors mentioned about biodegradation of silk-based materials -there need to be added information of degradation products and their activity, time of silk-based products stability, solutions for some problems depending on the type of silk-scaffolds used.
Reply: Your suggestion is very good, we may not have expanded the description when writing the manuscript, so we have added it in the corresponding section based on your suggestion (Please see: lines 77-91, 139-146).
Comment 3: In case of biological features – there is almost no information presented on the molecular level – this aspect needs to be improved by more detailed mechanism description (i.e., what about differentiation or redifferentaiation of chondrocytes, types of receptors, cells migration properties, transcription factors, enzymes activity?), as well as the quantitative data should be added (i.e. concentration/ dose, time of incubation, types of cells/issues). Additionally, the information is not clearly presented as obtained from in vitro or in vivo experiments – it obligatory needs to be rewritten. Please, add and comment the results from clinical studies.
Reply: Thank you for your professional advice. When we wrote the manuscript, the main description did not take this into account, and your suggestion is that our manuscript is more complete. We have made changes where appropriate (Please see: lines 196-206).
Comment 4: I strongly suggest to reread the manuscript carefully, since within the text very often are repeated general information (maybe because some parts were prepared by different coauthors) and information not related to cartilage tissue – it needs to be modified.
Reply: We are very sorry for the problem that has occurred in our manuscript, we have made changes to the article, thank you for your professional advice on our manuscript.
Comment 5: Please explain the sentence in lines 290-292, since silk fibroin is not a component of human extracellular matrix.
Reply: We're sorry for the problem with the statement due to our error, we've made changes, thank you very much for your professional advice (Please see: lines 315-316).
Comment 6: The drug -delivery chapter (4.2.3) seems to have more information about anti-cancer activity – please, reconsider this chapter modification.
Reply: Your suggestion is very important, and we have reorganized the content of the corresponding section (Please see: lines 383-396).
We hope that these responses would meet with your approval.
Once again, thank you very much for your effort in further improving this manuscript.

Round 2
Reviewer 1 Report
The authors have addressed all points of potential critisim and each suggestion made by the reviewer adequately and in detail.
Minor editing of English language required.
Author Response
Dear reviewer:
Thank you very much for accepting our manuscript “Silk Fibroin Based Biomaterials for Cartilage Regeneration in Osteoarthritis Therapy” (Manuscript ID: biomedicines-2465044). It is our pleasure to have the opportunity to publish papers in Biomedicines. Besides, Thank you very much for your kind letter and encouragement again, along with the comments from the reviewer and you concerning our manuscript “Silk Fibroin Based Biomaterials for Cartilage Regeneration in Osteoarthritis Therapy” (Manuscript ID: biomedicines-2465044). We have thoroughly considered all the comments and substantially revised our manuscript. The point-to-point answers and explanations for all comments were listed in separate pages following this letter.
Once again thank you for your kind consideration, and your efforts on this manuscript are greatly appreciated.
Best wishes to you!
Yours sincerely,
Prof. Li Chen and Ning Hu.
The detailed response is appended below.
Comment 1: Comments on the Quality of English Language
Minor editing of English language required.
Reply: Let me sincerely thank you for your professional review and your suggestions and recognition of this research work. We apologize for the deficiencies in our grammar, so we have re-polished the article and thank you again for your efforts on our manuscript.

Reviewer 3 Report
I have read the Authors’ response; they have answered some of my concerns and improved manuscript at some points. Still, within the text there is no correlation with Figures and Tables – no sentence presents any comment including data/mechanism presented in Figure xxx.
According to my comment about the presented pictures taken from other citied articles, despite the fact that the Authors included the copyright statement, I strongly suggest them to modify the pictures and present them differently – this will confirm the originality of the review submitted to Biomedicines journal, which is a journal with growing IF. For example, Figure 7 presents some experimental data obtained from ref. 144 not mentioned/commented in the manuscript. Another example - Figure 6 presents 3D fabrication of a scaffold made by bioprinting; in the manuscript there is no information about 3D bioprinting methods. Figure 5 should be modified – among the most interested is the photo of the material; 3D culture scheme and its grafting can be presented in other way be the authors. According to the Figure 4 – in the manuscript there is no mention about preparation of SF and nanosphere matrix (including usage of PVA). Other figures (1-3) are not so complicated that their “message” cannot be presented in other way. I would agree only for presentation of copyrighted material in case of the structure/morphology/photo of silk base biomaterials generated and produced by other scientific groups.
I asked Authors to read the text carefully, since it contains many repeats. Unfortunately it has not been read so carefully, since there are still some difficult fragments, i.e., lines 325-331 “Saha et al. used human bone marrow stromal cells (hBMSC) to assess the bone and/or cartilage induction capacity of mulberry and non-mulberry laminar silkin biomaterials in vitro in cartilage or osteoinduction medium for 4 to After 8 weeks, non-Mulberry constructs pre-inoculated with human bone marrow stromal cells exhibited prominent areas of new tissue containing chondrocyte like cells, while Mulberry constructs pre-inoculated with human bone marrow stromal cells formed bone-like nodules.” or lines 339-343 “Li et al. developed a silk The composite hydrogel of SF (SF) and carboxymethyl chitosan (CMCS) supported the adhesion, proliferation, glycosaminoglycan synthesis, and chondrogenic phenotype of rabbit articular chondrocytes, and the subcutaneous implantation of the hydrogel in mice showed no infection or local inflammatory response, indicating good biocompatibility in vivo [135].” These are only examples – therefore again I ask Authors to read the manuscript carefully – not as fragments, but look at it as a one whole thing!
Some other fragments do not allow to take the conclusion, i.e. what is culture 2 mentioned in lines 344-349?: “Liu et al. used electrostatic spinning to prepare a fibrillar SF (SF)/poly L-lactic acid (PLLA) scaffold, and chondrocytes spread well on the fibrous SF/PLLA scaffold and the secreted extracellular matrix, while in culture 2 showed low cytotoxicity to chondrocytes and good chondrocyte growth on the SF/PLLA scaffold after 1, 3, 5 and 7 days of direct contact, indicating good adhesion, biocompatibility and cytocompatibility of the SF/PLLA scaffold [ 136]. “
The above mentioned comments are only some examples, therefore, please, read the manuscript again and improve it.
Still, I do not feel satisfied with the presentation of silk based materials features, composition, types. There can be added as a table information of type of the scaffold used (composition and structure, i.e. fibre, sponge, microsphere; type of production (gel formation, 3D printing, etc) etc).
Overall, I suggest at least the major revision of the manuscript.
Author Response
Dear reviewer:
Thank you very much for accepting our manuscript “Silk Fibroin Based Biomaterials for Cartilage Regeneration in Osteoarthritis Therapy” (Manuscript ID: biomedicines-2465044). It is our pleasure to have the opportunity to publish papers in Biomedicines. Besides, Thank you very much for your kind letter and encouragement again, along with the comments from the reviewer and you concerning our manuscript “Silk Fibroin Based Biomaterials for Cartilage Regeneration in Osteoarthritis Therapy” (Manuscript ID: biomedicines-2465044). We have thoroughly considered all the comments and substantially revised our manuscript. The point-to-point answers and explanations for all comments were listed in separate pages following this letter.
Once again thank you for your kind consideration, and your efforts on this manuscript are greatly appreciated.
Best wishes to you!
Yours sincerely,
Prof. Li Chen and Ning Hu.
The detailed response is appended below.
Comment 1: I have read the Authors’ response; they have answered some of my concerns and improved manuscript at some points. Still, within the text there is no correlation with Figures and Tables – no sentence presents any comment including data/mechanism presented in Figure xxx.
According to my comment about the presented pictures taken from other citied articles, despite the fact that the Authors included the copyright statement, I strongly suggest them to modify the pictures and present them differently – this will confirm the originality of the review submitted to Biomedicines journal, which is a journal with growing IF. For example, Figure 7 presents some experimental data obtained from ref. 144 not mentioned/commented in the manuscript. Another example - Figure 6 presents 3D fabrication of a scaffold made by bioprinting; in the manuscript there is no information about 3D bioprinting methods. Figure 5 should be modified – among the most interested is the photo of the material; 3D culture scheme and its grafting can be presented in other way be the authors. According to the Figure 4 – in the manuscript there is no mention about preparation of SF and nanosphere matrix (including usage of PVA). Other figures (1-3) are not so complicated that their “message” cannot be presented in other way. I would agree only for presentation of copyrighted material in case of the structure/morphology/photo of silk base biomaterials generated and produced by other scientific groups.
Reply: Thank you very much for your valuable advice for my manuscript. We did quote many other pictures in the manuscript, and after listening to your suggestions, we redrew most of the pictures, but, as shown in Figures 1, 3, and 6, we felt that the pictures of other authors were very much in line with the content of our corresponding part during the painting process, so we cited them, and we confirmed that we had obtained the copyright. Other pictures and content have been modified according to your suggestions, thank you for your professional suggestions, hope to get your approval. (Please see: lines 95-99, 185-186, 240-241, 316-317, 347-350, 393-394).
Comment 2: I asked Authors to read the text carefully, since it contains many repeats. Unfortunately it has not been read so carefully, since there are still some difficult fragments, i.e., lines 325-331 “Saha et al. used human bone marrow stromal cells (hBMSC) to assess the bone and/or cartilage induction capacity of mulberry and non-mulberry laminar silkin biomaterials in vitro in cartilage or osteoinduction medium for 4 to After 8 weeks, non-Mulberry constructs pre-inoculated with human bone marrow stromal cells exhibited prominent areas of new tissue containing chondrocyte like cells, while Mulberry constructs pre-inoculated with human bone marrow stromal cells formed bone-like nodules.” or lines 339-343 “Li et al. developed a silk The composite hydrogel of SF (SF) and carboxymethyl chitosan (CMCS) supported the adhesion, proliferation, glycosaminoglycan synthesis, and chondrogenic phenotype of rabbit articular chondrocytes, and the subcutaneous implantation of the hydrogel in mice showed no infection or local inflammatory response, indicating good biocompatibility in vivo [135].” These are only examples – therefore again I ask Authors to read the manuscript carefully – not as fragments, but look at it as a one whole thing!
Some other fragments do not allow to take the conclusion, i.e. what is culture 2 mentioned in lines 344-349?: “Liu et al. used electrostatic spinning to prepare a fibrillar SF (SF)/poly L-lactic acid (PLLA) scaffold, and chondrocytes spread well on the fibrous SF/PLLA scaffold and the secreted extracellular matrix, while in culture 2 showed low cytotoxicity to chondrocytes and good chondrocyte growth on the SF/PLLA scaffold after 1, 3, 5 and 7 days of direct contact, indicating good adhesion, biocompatibility and cytocompatibility of the SF/PLLA scaffold [ 136]. “
Reply: We are very sorry for the problem due to our mistake, we have made changes in the article and modified the statements. (Please see: lines 255-264, 271-278, 285-287, 300-306, 330-343, 356-366).
Comment 3: Still, I do not feel satisfied with the presentation of silk based materials features, composition, types. There can be added as a table information of type of the scaffold used (composition and structure, i.e. fibre, sponge, microsphere; type of production (gel formation, 3D printing, etc) etc).
Reply: Your suggestion is very professional, for which I have added to the article, thank you again for your efforts for my manuscript (Please see: lines 154-184).
We hope that these responses would meet with your approval.
Once again, thank you very much for your effort in further improving this manuscript.
Round 3
Reviewer 3 Report
I have read the Authors response and the improved manucript. Overall, it can be accepted for publication (if the Editor has no problem with copied figures, despite the citation of their published sources - and this is the reaseon for the minor revision).
Author Response
Dear reviewer:
Thank you very much for accepting our manuscript “Silk Fibroin Based Biomaterials for Cartilage Regeneration in Osteoarthritis Therapy” (Manuscript ID: biomedicines-2465044). It is our pleasure to have the opportunity to publish papers in Biomedicines. Besides, thank you very much for your kind letter and encouragement again, along with the comments from the reviewer and you concerning our manuscript “Silk Fibroin Based Biomaterials for Cartilage Regeneration in Osteoarthritis Therapy” (Manuscript ID: biomedicines-2465044). We have thoroughly considered all the comments and substantially revised our manuscript. The point-to-point answers and explanations for all comments were listed in separate pages following this letter.
Once again thank you for your kind consideration, and your efforts on this manuscript are greatly appreciated.
Best wishes to you!
Yours sincerely,
Prof. Li Chen and Ning Hu.
The detailed response is appended below.
Comment 1: I have read the Authors response and the improved manucript. Overall, it can be accepted for publication (if the Editor has no problem with copied figures, despite the citation of their published sources - and this is the reaseon for the minor revision).
Reply: Thank you very much for your professional advice on the publication of our manuscript, and we really appreciate your help to us. We have determined to obtain the copyright of the figures that appear in the manuscript and have them properly referenced in the manuscript. At the same time, we have reworked the grammatical problems in the manuscript. Thank you again.
We hope that these responses would meet with your approval.
Once again, thank you very much for your effort in further improving this manuscript.